# Intramolecular Aminolactonization for Synthesis of Furoindolin-2-One

**DOI:** 10.3390/molecules27010102

**Published:** 2021-12-24

**Authors:** Kazuhiro Higuchi, Kazunori Matsumura, Takafumi Arai, Motoki Ito, Shigeo Sugiyama

**Affiliations:** Graduate School of Pharmaceutical Sciences, Meiji Pharmaceutical University, 2-522-1 Noshio, Kiyose, Tokyo 204-8588, Japan; m186217@alm.my-pharm.ac.jp (K.M.); y131009@alm.my-pharm.ac.jp (T.A.); mito@my-pharm.ac.jp (M.I.); sugiyama@my-pharm.ac.jp (S.S.)

**Keywords:** furoindolin-2-one, lapidilectine B, oxidative cyclization

## Abstract

Propellanes are polycyclic compounds in which tricyclic systems share one carbon–carbon single bond. Propellane frameworks that consist of larger sized rings are found in a variety of natural products. As an approach to the stereoselective synthesis of the propellane framework, one of the efficient methods is forming several rings in a single operation. Lapidilectine B (**1**) is composed of a propellane framework and was synthesized through the oxidative cyclization of trisubstituted alkenes. When the alkene with an ester moiety was treated with *N*-iodosuccinimide (NIS), iodocyclization proceeded to give the cyclic carbamate. On the other hand, when PhI(OAc)_2_ was allowed to react in the carboxyl form, a furoindolin-2-one structure corresponding to the A-B-C ring of lapidilectine B (**1**) was produced. Furthermore, when Pd(OAc)_2_ catalyst was used for cyclization under oxidative conditions, the product yield was improved.

## 1. Introduction

In 1992, lapidilectine B (**1**) (Figure 1) was isolated from the leaves of *Kopsia lapidilecta* Sleesen in Malaysia [1]. Biological activity of **1** shows a reverse multidrug resistance in vincristine-resistant KB cells [2].

The structural feature of **1** is possessing a unique propellane structure composed of indoline, γ-lactone, and azocane. In addition, the cyclohexane ring is connected to two pyrrolidine rings via two spirocenters. The first total synthesis of **1** was achieved by Pearson in 2001, where the Smalley cyclization was used to construct the framework of the B and C rings [3,4]. The optically active form of **1** was synthesized by Nishida in 2016 via desymmetrization of the spiro center using enantioselective deprotonation with a chiral lithium amide [5]. In 2018, Ma utilized manganese (III) mediated oxidative cyclization of 2-alkylindole to form the B–C–D rings of **1** in one operation [6]. We focused on the *anti*-configuration between oxygen and nitrogen atoms in the B–C–E propellane moiety, and postulated that the simultaneous functionalization of the alkene is a key reaction for ring construction in **1**.

As a methodology for double functionalization with heteroatoms on C-C double bond, Baran indicated the oxidative coupling reaction of *o*-iodoaniline and tryptamine using NIS to form the 3a substituted pyrroindoline motif in the synthesis of psychotrimine (Figure 1A) [7,8]. The electrophilic *N*-haloaniline and the nucleophilic carbamate of the tryptamine side chain were introduced into the 2,3-positions of the indole. One of promising methods for introducing of two heteroatoms onto an alkene with a transition metal catalyst is Sharpless’s asymmetric aminohydroxylation [9]. In 2007, Muñiz reported the palladium-catalyzed aminoalkoxylation of internal alkene bearing aniline and phenol moieties with PhI(OAc)_2_ (Figure 1B) [10,11]. Interestingly in this method, a furoindoline skeleton is assembled by closing two rings of a disubstituted alkene at once. To construct the 5-5 bicyclo B-C ring system in lapidilectine B (**1**), the *Z* trisubstituted alkene **2** was selected as the precursor for the ring closing reaction (Figure 1C). Here, we report the efficient synthesis of furoindolin-2-one **3** by the successive cyclization of trisubstituted alkene **2** with the goal of developing a method for synthesizing the polycyclic skeleton of **1**.

## 2. Results and Discussion

We synthesized trisubstituted alkenes **2** and **6** bearing an alkoxycarbonyl group on the aniline nitrogen (Figure 2). Aniline, its hydrochloride, and 4-oxopentanoic acid were condensed under heating at 200 °C to obtain lactam **4** according to the literature [12]. Thereafter, lactam **4** was transformed to imides **5a**–**e** by treating *n*-butyllithium with alkyl chloroformate or triethylamine with Boc_2_O, after which imides **5** were hydrolyzed with NaOH in THF-water to prepare carboxylic acids **2a**–**e**. Boc form **5c** was solvolyzed with sodium methoxide in MeOH to obtain methyl ester **6**.

First, we investigated a key cyclization using compounds **2c** and **6** bearing a Boc group (Figure 3). The treatment of carboxylic acid **2c** with NIS in a solution of MeCN and MeOH resulted in a complex mixture. On the other hand, when methyl ester **6** was treated under the same conditions as above, cyclic carbamate **7** was obtained in 89% yield as a single diastereomer. Iodocyclization triggered by the activation of the alkene with NIS, accompanied by the elimination of isobutylene from the Boc group, gave **7**. 

Trisubstituted alkene **2a** possessing a carboxylic acid moiety was then subjected to halogenating and oxidizing agents (Table 1). The reaction yielded many products when NIS was added to **2a** (entry 1), whereas the reaction did not proceed with NCS (entry 2). When **2a** was treated with CAN in MeCN solution, the trace amount of the desired cyclized product **3a** was produced (entry 3). When the reaction was carried out with PhI(OAc)_2_ under refluxing MeCN, product **3a** was obtained in 43% yield, along with many decomposition products (entry 4). Other hypervalent iodine reagents, PhI(OCOCF_3_)_2_, PhI(OH)OTs, and PhIO were tested, product **3a** was slightly detected on TLC, however substrate **2a** was mainly decomposed (entries 5–7). The fused ring structure of product **3a** was confirmed by the correlation between the methyl proton at position 8b and the 3a carbon in the HMBC experiment, and between the methyl proton at position 8b and the 3a proton in the nOe experiment.

A plausible reaction mechanism is shown in Figure 4. In conjunction with the oxidative activation of the aniline nitrogen atom in **2a** by PhI(OAc)_2_, the nitrogen atom undergoes nucleophilic attack by the π-electron of the alkene, after which the carboxylic acid cyclized to give furoindolin-2-one **3a**.

The conditions for palladium(II) mediated oxidative difunctionalization of alkenes reported by Muñiz were applied to trisubstituted alkene **2a** (Table 2). When 1 equivalent of the palladium reagent with 1.5 equivalents of PhI(OAc)_2_ treated with **2a** in DMF at room temperature, product **3a** was obtained in 19% yield with PdCl_2_, in 29% yield with Pd(OCOCF_3_)_2_, and in 58% yield with Pd(OAc)_2_ (entries 1–3). Using Pd(OAc)_2_ in MeCN, **3a** was obtained in 50% yield (entry 4). Based on the above results, a catalytic amount of Pd(OAc)_2_ was examined using DMF or MeCN. First, when 0.2 equivalent of Pd(OAc)_2_ was reacted at room temperature for 24 h, the yield of **3a** decreased to 14% in DMF and was maintained at 51% in MeCN (entries 5 and 6). When the reaction was performed under refluxing MeCN, the reaction time was shortened and the yield of **3a** increased to 78% (entry 7). Further reduction the amount of Pd(OAc)_2_ to 0.1 equivalent decreased the product yield to 47% (entry 8). With the optimum conditions in hand, the substituent effect on nitrogen atom was examined. The ethoxycarbonyl compound **2b** gave the corresponding product **3b** in 56% yield (entry 9). Even with **2c** possessing a bulky Boc group, product **3c** was obtained in 38% yield (entry 10). Even when the Cbz and Alloc substituents attached to nitrogen, products **3d** and **3e** were obtained in 33% and 29% yields, respectively (entries 11 and 12).

A reaction mechanism for the cyclization was proposed based on the report by Muñiz. [10,13,14,15,16] (Figure 5). First, Pd(OAc)_2_ reacts with the nitrogen atom of substrate **2a** and coordinates with the alkene to form intermediate **8**. Subsequently, *anti*-oxypalladation occurs on the alkene to generate the six-membered palladacycle intermediate **9**. Furthermore, palladium(II) is oxidized to palladium(IV) by PhI(OAc)_2_ to form intermediate **10**, followed by the reductive elimination of palladium(IV) in **10** to produce **3a** and palladium(II) species.

## 3. Materials and Methods

### 3.1. General Information

All melting points were measured on a Yanagimoto micro melting point apparatus. IR spectra were recorded on a JASCO FT/IR-4100 spectrometer and absorbance bands are reported in wavenumber (cm^−1^). ^1^H NMR spectra were recorded on JEOL JNM-AL 300 and 400 (300 and 400 MHz) spectrometer or JEOL JNM-ECS 400 (400 MHz) spectrometer. Chemical shifts are reported relative to internal standard (tetramethylsilane at δ_H_ 0.00, CDCl_3_ at δ_H_ 7.26). Data are presented as follows: chemical shift (δ, ppm), multiplicity (s = singlet, d = doublet, t = triplet, q = quartet, m = multiplet), coupling constant and integration. ^13^C NMR spectra were recorded on JEOL JNM-ECA 400 (100 MHz) spectrometer. Chemical shifts are reported relative to internal standard (CDCl_3_ at δ 77.00). Mass spectra were recorded on a JEOL JMS 700 instrument with a direct inlet system. Column chromatography was carried out on Kanto silica gel 60 N (40–50 mesh). Analytical thin layer chromatography (TLC) was carried out on Merck Kieselgel 60 F_254_ plates with visualization by ultraviolet, anisaldehyde stain solution or phosphomolybdic acid stain solution. All non-aqueous reactions were carried out in flame-dried glassware under argon atmosphere unless otherwise noted. Reagents were purchased from TCI, nacalai tesque, Kanto chemical, FUJIFILM Wako chemicals or Sigma-Aldrich. Reagents and solvents were used without purification. Substrate **4** was synthesized by a reported method [12], and spectra data agreed with the literature values.

### 3.2. Preparation and Characterization of Novel Compounds

#### 3.2.1. General Procedure for the Synthesis of Alkyl 5-Methyl-2-oxo-2,3-dihydro-1*H*-benzo[*b*]azepine-1-carboxylate (**5a**,**b**,**d**,**e**)

A solution of **4** in THF was cooled to −78 °C, *n*-BuLi (1.6 M in *n*-hexane) was added, and the mixture was stirred for 30 min. Alkyl chloroformate was added to the mixture, and the mixture was stirred for 20 min at same temperature. After confirming the disappearance of substrate **4** by TLC, saturated the NH_4_Cl aqueous solution was added. The reaction mixture was extracted with CH_2_Cl_2_. The organic layer was washed with a saturated aqueous NaCl solution, dried with anhydrous MgSO_4_, filtered. The filtrate was concentrated under reduced pressure. The residue was purified by silica gel column chromatography to give **5**.

##### Methyl 5-Methyl-2-oxo-2,3-dihydro-1*H*-benzo[*b*]azepine-1-carboxylate (**5a**)

**4** (0.10 g, 0.58 mmol), THF (2.3 mL), *n*-BuLi (0.73 mL, 1.2 mmol), methyl chloroformate (90 μL, 1.2 mmol), eluent (AcOEt/*n*-hexane = 1/2), **5a** (0.14 g, quant.). Pale yellow oil. IR (KBr): 3547, 2954, 1779, 1224 cm^−1^. ^1^H-NMR (400 MHz, CDCl_3_): δ 2.22 (3H, s), 2.69 (1H, brs), 3.04 (1H, brs), 3.79 (3H s), 5.90 (1H, tq, *J* = 7.1, 1.4 Hz), 7.24–7.27 (1H, m), 7.30–7.37 (2H, m), 7.47–7.49 (1H, m). ^13^C-NMR (100 MHz, CDCl_3_): δ 20.8, 37.4, 54.1, 121.6, 126.5, 127.0, 127.2, 127.6, 135.5, 135.9, 136.4, 153.9, 172.5. MS (EI): *m*/*z* (%) 231 (M^+^, 13), 190 (11), 189 (100), 144 (37). HRMS (EI): **m*/*z** calcd for C_13_H_13_NO_3_ [M^+^] 231.0895, found 231.0892.

##### Ethyl 5-Methyl-2-oxo-2,3-dihydro-1*H*-benzo[*b*]azepine-1-carboxylate (**5b**)

**4** (0.13 g, 0.74 mmol), THF (7.4 mL), *n*-BuLi (0.69 mL, 1.1 mmol), ethyl chloroformate (0.14 mL, 1.5 mmol), eluent (AcOEt/*n*-hexane = 1/2), **5b** (0.13 g, 70%). Pale yellow oil. IR (KBr): 3381, 2981, 1777, 1219 cm^−1^. ^1^H-NMR (300 MHz, CDCl_3_): δ 1.25 (3H, t, *J* = 7.1 Hz), 2.21 (3H s), 2.69 (1H, brs), 3.02 (1H, brs), 4.25 (2H, brq, *J* = 6.2 Hz), 5.90 (1H, tq, *J* = 7.0, 1.5 Hz), 7.21–7.34 (3H, m), 7.46–7.49 (1H, m). ^13^C-NMR (100 MHz, CDCl_3_): δ 14.4, 25.0, 34.3, 61.2, 115.2, 119.5, 121.4, 123.3, 128.0, 128.2, 134.4, 137.9, 153.9, 177.6. MS (EI): **m*/*z** (%) 245 (M^+^, 16), 204 (12), 203 (100), 173 (10), 172 (17), 158 (14), 144 (55), 143 (12), 131 (19), 130 (27). HRMS (EI): **m*/*z** calcd for C_14_H_15_NO_3_ [M^+^] 245.1052, found 245.1050.

##### *tert*-Butyl 5-Methyl-2-oxo-2,3-dihydro-1*H*-benzo[*b*]azepine-1-carboxylate (**5c**)

A solution of **4** (0.31 g, 1.8 mmol) in CH_2_Cl_2_ (3.0 mL) was added Boc_2_O (0.62 mL, 2.7 mmol), DMAP (44 mg, 0.36 mmol) and Et_3_N (0.75 mL, 5.4 mmol) and stirred at rt for 24 h. After confirming the consumption of substrate **4** by TLC, saturated aqueous solution of NaHCO_3_ was added to the reaction mixture. After extraction with CH_2_Cl_2_, the organic layer was washed with a saturated aqueous NaCl solution, dried with anhydrous MgSO_4_, filtered, and the filtrate was concentrated under reduced pressure. The obtained residue was purified by silica gel column chromatography (AcOEt/*n*-hexane = 1/2) to obtain **5c** (0.44 g, 90%) as pale yellow oil. IR (KBr): 3374, 2979, 2253, 1774, 1240 cm^−1^. ^1^H-NMR (400 MHz, CDCl_3_): δ 1.44 (9H, s), 2.21 (3H, s), 2.66 (1H, brs), 2.98 (1H, brs), 5.90 (1H, tq, *J* = 6.8, 1.4 Hz), 7.22–7.31 (3H, m), 7.45–7.47 (1H, m). ^13^C-NMR (100 MHz, CDCl_3_): δ 20.9, 27.6 (3C), 37.2, 83.6, 121.8, 126.4, 126.5, 126.6, 127.3, 135.1, 136.1 136.3, 151.9, 171.7. MS (EI): **m*/*z** (%) 273 (M^+^, 7), 174 (12), 173 (100), 172 (14), 158 (18), 144 (23), 131 (34), 130 (17), 57 (33). HRMS (EI): **m*/*z** calcd for C_16_H_19_NO_3_ [M^+^] 273.1365, found 273.1370.

##### Benzyl 5-Methyl-2-oxo-2,3-dihydro-1*H*-benzo[*b*]azepine-1-carboxylate (**5d**)

**4** (0.19 g, 1.1 mmol), THF (11 mL), *n*-BuLi (1.0 mL, 1.6 mmol), benzyl chloroformate (0.84 mL, 2.2 mmol), eluent (AcOEt/*n*-hexane = 1/2), **5d** (0.31 g, 92%). Pale yellow oil. IR (KBr): 3535, 2949, 1779, 1218 cm^−1^. ^1^H-NMR (400 MHz, CDCl_3_): δ 2.20 (3H, s), 2.69 (1H, brs), 3.04 (1H, brs), 5.16 (1H, brs), 5.25 (1H, brs), 5.89 (1H, tq, *J* = 6.9, 1.4 Hz), 7.15 (1H, dd, *J* = 8.0, 1.3 Hz), 7.14–7.37 (7H, m), 7.46 (1H, dd, *J* = 7.8, 1.6 Hz). ^13^C-NMR (100 MHz, CDCl_3_): δ 20.8, 37.4, 68.6, 121.5, 126.5, 126.9, 127.0, 127.3, 127.4, 128.0, 128.2, 128.4 (2C), 134.9, 135.5, 135.8, 136.4, 153.1, 172.4. MS (EI): **m*/*z** (%) 307 (M^+^, 14), 266 (11), 265 (59), 221 (48), 172 (24), 157 (18), 144 (15), 91 (100). HRMS (EI): **m*/*z** calcd for C_19_H_17_NO_3_ [M^+^] 307.1208, found 307.1206.

##### Allyl 5-Methyl-2-oxo-2,3-dihydro-1*H*-benzo[*b*]azepine-1-carboxylate (**5e**)

**4** (0.10 g, 0.58 mmol), THF (5.8 mL), *n*-BuLi (0.54 mL, 0.87 mmol), allyl chloroformate (0.12 mL, 1.2 mmol), eluent (AcOEt/*n*-hexane = 1/2), **5e** (0.12 g, 78%). Pale yellow oil. IR (KBr): 3370, 2979, 1773, 1240 cm^−1^. ^1^H-NMR (300 MHz, CDCl_3_): δ 2.21 (3H, s), 2.70 (1H, brs), 3.03 (1H, brs), 4.67 (2H, brs), 5.18–5.29 (2H, m), 5.80–5.93 (2H, m), 7.25–7.37 (3H, m), 7.45–7.49 (1H, m). ^13^C-NMR (100 MHz, CDCl_3_): δ 25.1, 34.6, 65.9, 118.1, 119.8, 121.7, 123.6, 128.1, 128.2, 132.4, 134.3, 137.7, 153.6, 162.1, 177.8. MS (EI): **m*/*z** (%) 257 (M^+^, 16), 216 (14), 215 (100), 184 (11), 172 (19), 171 (10), 170 (21), 156 (12), 144 (35), 143 (12), 130 (19). HRMS (EI): **m*/*z** calcd for C_15_H_15_NO_3_ [M^+^] 257.1052, found 257.1052.

#### 3.2.2. General Procedure for the Synthesis of (*Z*)-4-{2-[(Alkoxycarbonyl)amino]phenyl}pent-3-enoic acids **2**

A solution of **5** in THF and H_2_O was added NaOH and stirred at room temparature for 2 h. After confirming the consumption of substrate **5** by TLC, the reaction mixture was acidified with 10% HCl and extracted with AcOEt. The organic layer was washed with brine, dried with anhydrous MgSO_4_, filtered. The filtrate was concentrated under reduced pressure to obtain **2**.

##### (*Z*)-4-{2-[(Methoxycarbonyl)amino]phenyl}pent-3-enoic Acid (**2a**)

**5a** (0.11 g, 0.49 mmol), THF (4.9 mL), H_2_O (4.9 mL), NaOH (58 mg, 2.5 mmol), **2a** (0.15 g, quant.). Yellow oil. IR (KBr): 3311, 2959, 1710, 1523, 1219 cm^−1^. ^1^H-NMR (500 MHz, CDCl_3_): δ 2.00 (3H, s), 2.85 (2H, d, *J* = 7.3 Hz), 3.73 (3H, s), 5.78 (1H, td, *J* = 7.3, 1.2 Hz), 6.98–7.00 (2H, m), 7.06 (1H, t, *J* = 7.3 Hz), 7.28 (1H, t, *J* = 8.8 Hz), 8.02 (1H, brs). ^13^C-NMR (125 MHz, CDCl_3_): δ 25.0, 34.2, 52.2, 119.6, 121.4, 123.5, 128.0, 128.3, 130.1, 134.4, 138.1, 154.2, 177.2. MS (FAB): **m*/*z** (%) 250 (M+H^+^, 64), 204 (11). HRMS (FAB): **m*/*z** calcd for C_13_H_16_NO_4_ [M+H^+^] 250.1079, found 250.1078.

##### (*Z*)-4-{2-[(Ethoxycarbonyl)amino]phenyl}pent-3-enoic Acid (**2b**)

**5b** (0.23 g, 0.92 mmol), THF (9.2 mL), H_2_O (9.2 mL), NaOH (0.18 g, 4.6 mmol), **2b** (0.23 g, 95%). Pale yellow oil. IR (KBr): 3396, 2979, 1731, 1215 cm^−1^. ^1^H-NMR (500 MHz, CDCl_3_): δ 1.27 (3H, t, *J* = 7.3 Hz), 1.99 (3H s), 2.83 (2H, d, *J* = 6.4 Hz), 4.18 (2H, q, *J* = 7.0 Hz), 5.77 (1H, brs), 6.98–7.06 (2H, m), 7.27 (1H, t, *J* = 7.0 Hz), 8.02 (1H, brs). ^13^C-NMR (125 MHz, CDCl_3_): δ 14.4, 25.1, 34.5, 61.3, 119.6, 121.6, 123.4, 128.1, 128.2, 130.2, 134.4, 137.8, 154.0, 177.8. MS (EI): **m*/*z** (%) 263 (M^+^, 15), 204 (22), 203 (100), 144 (69), 132 (11), 131 (21), 130 (31). HRMS (EI): **m*/*z** calcd for C_14_H_17_NO_4_ [M^+^] 263.1158, found 263.1154.

##### (*Z*)-4-{2-[(*tert*-Butoxycarbonyl)amino]phenyl}pent-3-enoic Acid (**2c**)

**5c** (0.27 g, 0.97 mmol), THF (9.7 mL), H_2_O (9.7 mL), NaOH (0.20 g, 4.9 mmol), **2c** (0.28 g, 99%). Yellow solids. mp: 126–128 °C, IR (KBr): 3410, 2977, 1730, 1164 cm^−1^. ^1^H-NMR (300 MHz, CDCl_3_): δ 1.49 (9H, s), 2.01 (3H, s), 2.86 (2H, d, *J* = 6.2 Hz), 5.79 (1H, td, *J* = 7.3, 1.4 Hz), 6.71 (1H, s), 6.96–7.05 (2H, m), 7.23–7.29 (1H, m), 8.01 (1H, d, *J* = 8.0 Hz). ^13^C-NMR (100 MHz, CDCl_3_): δ 25.0, 28.2 (3C), 34.3, 80.5, 119.6, 121.3, 123.0, 128.0, 128.2, 129.8, 134.8, 138.0, 153.0, 177.4. MS (FAB): **m*/*z** (%) 292 (M+H^+^, 42), 236 (68), 192 (45), 146 (23). HRMS (FAB): **m*/*z** calcd for C_16_H_22_NO_4_ [M+H^+^] 292.1549, found 292.1544.

##### (*Z*)-4-{2-[(Benzyloxycarbonyl)amino]phenyl}pent-3-enoic Acid (**2d**)

**5d** (0.12 g, 0.37 mmol), THF (3.7 mL), H_2_O (3.7 mL), NaOH (74 mg, 1.9 mmol), **2d** (0.13 g, quant.). Pale yellow oil. IR (KBr): 3401, 2965, 1736, 1216 cm^−1^. ^1^H-NMR (500 MHz, CDCl_3_): δ 1.98 (3H, s), 2.81 (2H, d, *J* = 7.0 Hz), 5.16 (2H, brd, *J* = 9.5 Hz), 5.74 (1H, qt, *J* = 1.5, 8.0 Hz), 6.98 (1H, d, *J* = 7.0 Hz), 7.05 (1H, t, *J* = 7.3 Hz), 7.25–7.39 (6H, m), 8.06 (1H, brs). ^13^C-NMR (125 MHz, CDCl_3_): δ 25.1, 34.3, 66.9, 119.7, 121.4, 123.5, 127.0, 128.1, 128.2, 128.3 (2C), 128.5 (2C), 130.1, 134.3, 136.1, 137.9, 153.5, 177.6. MS (EI): **m*/*z** (%) 325 (M^+^, 10), 265 (12), 190 (40), 175 (10), 144 (54), 130 (14), 91 (100). HRMS (EI): **m*/*z** calcd for C_19_H_19_NO_4_ [M^+^] 325.1314, found 325.1312.

##### (*Z*)-4-{2-[(Allyloxycarbonyl)amino]phenyl}pent-3-enoic Acid (**2e**)

**5e** (0.18 g, 0.65 mmol), THF (6.5 mL), H_2_O (6.5 mL), NaOH (0.13 g, 3.3 mmol), **2e** (0.14 g, 79%). Pale yellow oil. IR (KBr): 3393, 2940, 1732, 1213 cm^−1^. ^1^H-NMR (500 MHz, CDCl_3_): δ 1.98 (3H, s), 2.82 (2H, brs), 4.63 (2H, d, *J* = 4.5 Hz), 5.21 (1H, d, *J* = 10.5 Hz), 5.32 (1H, d, *J* = 17.0 Hz), 5.76 (1H, brs), 5.89–5.98 (1H, m), 6.96–7.08 (2H, m), 7.25–7.28 (1H, m), 8.01 (1H, brs). ^13^C-NMR (125 MHz, CDCl_3_): δ 25.1, 34.7, 65.9, 118.1, 119.7, 121.8, 123.6, 128.1, 128.2, 130.4, 132.5, 134.3, 137.6, 153.6, 177.8. MS (EI): **m*/*z** (%) 275 (M^+^, 21), 216 (27), 215 (100), 190 (21), 172 (15), 170 (11), 145 (11), 144 (86), 131 (21), 130 (40). HRMS (EI): **m*/*z** calcd for C_15_H_17_NO_4_ [M^+^] 275.1158, found 275.1159.

##### Methyl (*Z*)-4-{2-[(*tert*-Butoxycarbonyl)amino]phenyl}pent-3-enoate (**6**)

A solution of **5c** (0.27 g, 1.0 mmol) in MeOH (10 mL) was added NaOMe (54 mg, 5.0 mmol) and stirred at rt for 1.5 h. After confirming the consumption of substrate **5c** by TLC, the reaction mixture was acidified with 10% HCl and extracted with AcOEt, the organic layer was washed with brine, dried with anhydrous MgSO_4_, filtered, and the filtrate was concentrated under reduced pressure to obtain **6** (0.28 g, 93%). Colorless oil. IR (KBr): 3399, 2978, 1731, 1516, 1160 cm^−1^. ^1^H-NMR (500 MHz, CDCl_3_): δ 1.51 (9H, s), 2.01 (3H, s), 2.82 (2H, d, *J* = 7.5 Hz), 3.65 (3H, s), 5.81 (1H, t, *J* = 7.5 Hz), 6.72 (1H, brs), 6.96–7.04 (2H, m), 7.25 (1H, t, *J* = 8.0 Hz), 8.03 (1H, d, *J* = 8.0 Hz). ^13^C-NMR (125 MHz, CDCl_3_): δ 25.0, 28.3 (3C), 34.4, 51.9, 80.3, 119.5, 121.9, 123.0, 128.0, 128.1, 129.9, 134.8, 137.5, 153.0, 172.2. MS (FAB): **m*/*z** (%) 306 (M+H^+^, 45), 250 (69), 206 (100), 146 (29), 131 (27). HRMS (FAB): **m*/*z** calcd for C_17_H_24_NO_4_ [M+H^+^] 306.1705, found 306.1712.

#### 3.2.3. Methyl 3-Iodo-3-(4-methyl-2-oxo-1,4-dihydro-2H-benzo[d][1,3]oxazin-4-yl)propanoate (**7**)

To a solution of **6** (30 mg, 0.10 mmol) in a mixture of MeCN/MeOH (1.2 mL, 20/1) was added a solution of NIS (3.4 mg, 0.15 mmol) in MeCN (0.30 mL) at −45 °C. The reaction mixture was stirred and allowed to room temperature for 5 h. After confirming the consumption of substrate **6** by TLC, the reaction was quenched with aqueous solution of Na_2_S_2_O_3_. After extraction with AcOEt, the organic layer was washed with brine, dried with anhydrous MgSO_4_, filtered. The filtrate was concentrated under reduced pressure. The residue was purified by silica gel column chromatography (AcOEt/*n*-hexane = 1/2) to obtain **7** (33 mg, 89%) as a pale yellow oil. IR (KBr): 3619, 3019, 1730, 1046 cm^−1^. ^1^H-NMR (400 MHz, CDCl_3_): δ 1.87 (3H, s), 3.11 (1H, dd, *J* = 16.5, 10.2 Hz), 3.20 (1H, dd, *J* = 16.6, 3.9 Hz), 3.70 (3H, s), 4.79 (1H, dd, *J* = 10.2, 3.6 Hz), 6.91 (1H d, *J* = 7.8 Hz), 7.08 (1H, t, *J* = 7.8 Hz), 7.17 (1H, d, *J* = 7.3 Hz), 7.30 (1H, t, *J* = 7.8 Hz), 9.21 (1H, s). ^13^C-NMR (100 MHz, CDCl_3_): δ 24.5, 33.8, 40.9, 52.3, 85.5, 115.1, 122.2, 123.6, 124.4, 129.9, 133.9, 151.0, 170.7. MS (EI): **m*/*z** (%) 375 (M^+^, 13), 162 (100), 144 (27), 130 (12). HRMS (EI): **m*/*z** calcd for C_13_H_14_NO_4_I [M^+^] 374.9968, found 374.9960.

#### 3.2.4. General Procedure for the Oxidative Cyclization of **2**

To a solution of **2** in MeCN was added Pd(OAc)_2_ and PhI(OAc)_2_. The reaction mixture was stirred under reflux for 6 h. After confirming the consumption of substrate **2** by TLC, the reaction was quenched with aqueous solution of Na_2_S_2_O_3_. After extraction with AcOEt, the organic layer was washed with brine, dried with anhydrous MgSO_4_, filtered. The filtrate was concentrated under reduced pressure. The residue was purified by silica gel column chromatography to obtain **3**.

##### Methyl (3a*R**,8b*R**)-8b-Methyl-2-oxo-2,3,3a,8b-tetrahydro-4*H*-furo[3,2-*b*]indole-4-carboxylate (**3a**)

**2a** (78 mg, 0.31 mmol), MeCN (3.1 mL), Pd(OAc)_2_ (14 mg, 60 μmol), PhI(OAc)_2_ (0.15 g, 0.47 mmol), eluent (AcOEt/*n*-hexane = 1/2), **3a** (59 mg, 78%). Yellow solid. mp: 160–162 °C. IR (KBr): 3021, 1776, 1716 cm^−1^. ^1^H-NMR (400 MHz, CDCl_3_): δ 1.82 (3H, s), 2.96 (1H, d, *J* = 19.5 Hz), 3.23 (1H, dd, *J* = 19.5, 8.2 Hz), 3.89 (3H s), 4.60 (1H, d, *J* = 6.3 Hz), 7.11 (1H, t, *J* = 7.3 Hz), 7.36 (1H, d, *J* = 7.8 Hz), 7.40 (1H, d, *J* = 7.8 Hz), 7.91 (1H, brs). ^13^C-NMR (100 MHz, CDCl_3_): δ 24.4, 36.3, 53.1, 64.6, 89.4, 115.2, 123.7, 124.1, 129.8, 131.1, 141.8, 152.6, 174.1. MS (EI): **m*/*z** (%) 247 (M^+^, 83), 202 (60), 188 (100), 158 (35), 146 (17), 144 (83), 143 (21). HRMS (EI): **m*/*z** calcd for C_13_H_13_NO_4_ [M^+^] 247.0845, found 247.0847.

##### Ethyl (3a*R**,8b*R**)-8b-Methyl-2-oxo-2,3,3a,8b-tetrahydro-4*H*-furo[3,2-*b*]indole-4-carboxylate (**3b**)

**2b** (0.23 g, 0.87 mmol), Pd(OAc)_2_ (40 mg, 0.18 mmol), PhI(OAc)_2_ (0.42 g, 1.3 mmol), MeCN (8.7 mL), eluent (AcOEt/*n*-hexane = 1/2), **3b** (0.10 g, 56%). Yellow solid. mp: 116–118 °C. IR (KBr): 3532, 2980, 1778, 1713, 1225 cm^−1^. ^1^H-NMR (400 MHz, CDCl_3_): δ 1.38 (3H, brs), 1.83 (3H, s), 2.98 (1H, d, *J* = 19.0 Hz), 3.23 (1H, dd, *J* = 19.2, 8.2 Hz), 4.35 (2H, brq, *J* = 6.6 Hz), 4.60 (1H, dd, *J* = 8.2, 2.0 Hz), 7.10 (1H, t, *J* = 7.3 Hz), 7.36 (1H, d, *J* = 7.7 Hz), 7.41 (1H, d, *J* = 8.2 Hz), 7.93 (1H, brs). ^13^C-NMR (100 MHz, CDCl_3_): δ 14.5, 24.4, 36.5, 62.2, 64.6, 89.4, 115.2, 123.6, 124.1, 130.0, 131.1, 141.7, 152.2, 174.3. MS (EI): **m*/*z** (%) 261 (M^+^, 71), 188 (42), 146 (23), 144 (100), 130 (30). HRMS (EI): **m*/*z** calcd for C_14_H_15_NO_4_ [M^+^] 261.1001, found 261.0999.

##### *tert*-Butyl (3a*R**,8b*R**)-8b-Methyl-2-oxo-2,3,3a,8b-tetrahydro-4*H*-furo[3,2-*b*]indole-4-carboxylate (**3c**)

**2c** (0.16 g, 0.55 mmol), Pd(OAc)_2_ (25 mg, 0.11 mmol), PhI(OAc)_2_ (0.27 g, 0.83 mmol), MeCN (5.5 mL), eluent (AcOEt/*n*-hexane = 1/2), **3c** (60 mg, 38%). Yellow solid. mp: 172–174 °C. IR (KBr): 3534, 2976, 1778, 1711, 1381, 1166 cm^−1^. ^1^H-NMR (500 MHz, CDCl_3_): δ 1.58 (9H, s), 1.82 (3H, s), 2.99 (1H, brs), 3.20 (1H, dd, *J* = 19.5, 8.0 Hz), 4.54 (1H, brs), 7.08 (1H, t, *J* = 7.5 Hz), 7.35 (1H, t, *J* = 8.0 Hz), 7.39 (1H, d, *J* = 7.5 Hz), 7.92 (1H, brs). ^13^C-NMR (100 MHz, CDCl_3_): δ 24.4, 28.3 (3C), 36.7, 64.6, 82.1, 89.5, 115.1, 123.3, 124.1, 129.6, 131.0, 142.3, 152.1, 174.5. MS (EI): **m*/*z** (%) 289 (M^+^, 27), 233 (100), 189 (41), 188 (23), 144 (58), 57 (68). HRMS (EI): **m*/*z** calcd for C_16_H_19_NO_4_ [M^+^] 289.1314, found 289.1311.

##### Benzyl (3a*R**,8b*R**)-8b-Methyl-2-oxo-2,3,3a,8b-tetrahydro-4*H*-furo[3,2-*b*]indole-4-carboxylate (**3d**)

**2d** (0.28 g, 0.85 mmol), Pd(OAc)_2_ (38 mg, 0.17 mmol), PhI(OAc)_2_ (0.41 g, 1.3 mmol), MeCN (8.5 mL)), eluent (AcOEt/*n*-hexane = 1/2), **3d** (90 mg, 33%). Yellow solid. mp: 140–142 °C. IR (KBr): 3532, 2974, 1777, 1719, 1402, 1226 cm^−1^. ^1^H-NMR (500 MHz, CDCl_3_): δ 1.81 (3H, s), 2.93 (1H, brs), 3.17 (1H, brs), 4.60 (1H, brs), 5.27 (1H, brs), 5.31 (1H, brs), 7.10 (1H, brs), 7.37-7.41 (7H, m), 7.96 (1H, brs). ^13^C-NMR (100 MHz, CDCl_3_): δ 24.5, 36.6, 64.6, 67.9, 89.6, 115.3, 123.8, 124.1, 128.3 (2C), 128.6, 128.8 (2C), 129.8, 131.1, 135.4, 141.7, 151.8, 174.1. MS (EI): **m*/*z** (%) 323 (M^+^, 30), 279 (16), 144 (18), 91 (100). HRMS (EI): **m*/*z** calcd for C_19_H_17_NO_4_ [M^+^] 323.1158, found 323.1155.

##### Allyl (3a*R**,8b*R**)-8b-Methyl-2-oxo-2,3,3a,8b-tetrahydro-4*H*-furo[3,2-*b*]indole-4-carboxylate (**3e**)

**2e** (0.14 g, 0.52 mmol), Pd(OAc)_2_ (22 mg, 0.10 mmol), PhI(OAc)_2_ (0.25 g, 0.78 mmol), MeCN (5.2 mL), eluent (AcOEt/*n*-hexane = 1/2), **3e** (41 mg, 29%). Yellow solid. mp: 70–72 °C. IR (KBr): 3533, 2933, 1778, 1720, 1397, 1226 cm^−1^. ^1^H-NMR (300 MHz, CDCl_3_): δ 1.83 (3H, s), 2.98 (1H, d, *J* = 18.0 Hz), 3.24 (1H, dd, *J* = 18.0, 8.1 Hz), 4.62 (1H dd, *J* = 8.1, 2.7 Hz), 4.77 (2H, brs), 5.32 (1H, d, *J* = 10.5 Hz), 5.39 (1H, d, *J* = 17.1 Hz), 5.95–6.05 (1H, m), 7.11 (1H, t, *J* = 7.5 Hz), 7.28–7.43 (2H, m), 7.90 (1H, brs). ^13^C-NMR (125 MHz, CDCl_3_, major): δ 24.5, 36.6, 64.6, 66.7, 89.5, 115.3, 119.1, 123.8, 124.1, 129.8, 131.2, 131.8, 141.9, 151.7, 174.2. MS (EI): **m*/*z** (%) 273 (M^+^, 100), 188 (22), 146 (56), 144 (85), 41 (20). HRMS (EI): **m*/*z** calcd for C_15_H_15_NO_4_ [M^+^] 273.1001, found 273.1000.

## 4. Conclusions

We investigated the construction of the furoindolin-2-one structure corresponding to the A to C ring system of lapidilectine B (**1**). The furoindolin-2-one structure was constructed directly by double functionalization of the internal alkene with carboxyl and carbamoyl groups under oxidative conditions using PhI(OAc)_2_. Furthermore, we succeeded in obtaining the desired product in higher yield with a palladium(II) catalyst under oxidative conditions. We are currently synthesizing more advanced substrates for the synthesis of lapidilectine B (**1**).

## Data Availability

The data presented in this study are available in this article.

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
