# Peer review of "Intramolecular Aminolactonization for Synthesis of Furoindolin-2-One"

_molecules, 2021, doi:10.3390/molecules27010102_

Round 1
Reviewer 1 Report
Dear authors
The manuscript should be accompanied by supporting information with NMR spectra. Without such information I cannot assess the manuscript properly.
Author Response
Responce to comment of reviewer 1
We added NMR spectra as supplementary materials of obtained compounds including the HMBC and nOe experiments of compound 3a. Please see the attachment file.

Reviewer 2 Report
ч
Kazuhiro Higuchi et al. report the efficient synthesis of furoindolin-2-one by the successive cyclization of trisubstituted alkene with the goal of developing a method for synthesizing the polycyclic skeleton of Lapidilectine B. It should be noted that, Biological activity of lapidilectine B shows a reverse multidrug resistance in vincristineesistant KB cells. In the beginning were synthesized trisubstituted alkenes bearing an alkoxycarbonyl group on the aniline nitrogen. The treatment of a trisubstituted alkene with an alkoxycarbonyl group on aniline nitrogen with a free carboxyl group using NIS in a solution of MeCN and MeOH leads to a complex mixture. Whereas cyclization of a trisubstituted alkene bearing an alkoxycarbonyl group on aniline nitrogen with an additional methyl ester group gives a cyclic carbamate. Oxidative cyclization of an alkene trisubstituted alkene with an alkoxycarbonyl group on aniline nitrogen with a free carboxyl group using other reagents based on hypervalent iodine also did not give a high yield of the cyclization product.
Despite the not high novelty of the study, the article can be published in a journal, since the demonstrated the construction of the furoindolin-2-one structure corresponding to 334 the A to C ring system of lapidilectine B (1). The furoindolin-2-one structure was con-335 structed directly by double functionalization of the internal alkene with carboxyl and car-336 bamoyl groups under oxidative conditions using PhI(OAc)2. Furthermore, authors succeeded 337 in obtaining the desired product in higher yield with a palladium(II) catalyst under oxi-338 dative conditions.

Author Response
To response to the comment of reviewer 2,
Thank you for your kindy comment.
Reviewer 3 Report
The manuscript entitled “Intramolecular Aminolactonization for Synthesis of Furoindolin-2-one” by K. Higuchi et al. described the furoindolin-2-one structure synthesis by double functionalization of the internal alkene with carboxyl and carbamoyl groups under oxidative conditions using PhI(OAc)2. The manuscript may be of general interest to the researchers of this field, but the manuscript lacks some information that the author should consider and incorporate in the present form of the manuscript. Here are a few concerns that need to be addressed in the present form of the manuscript.
- Authors should once again carefully check the abstract following the style of structured abstract. It should be added more background addressed in a broad context and highlight the purpose of the study.
- It should be added an information of commercial suppliers.
- Authors should add supplementary materials of obtained compounds, especially for confirmation of the correlation between the methyl proton at position 8b and the 3a carbon in the HMBC experiment, and between the methyl proton at position 8b and the 3a proton in the nOe experiment.
- Compounds 2a-e in the experimental part are Z-isomers. It should be explain the formation of such isomer in “Results and Discussion”.
Author Response
To response to the comment of reviewer 3,
1. We check the abstract following the style of structured abstract and added background addressed in a context and highlight the purpose of the study.
2. We added next sentence at 3.1. General information. "Reagents were purchased from TCI, nacalai tesque, Kanto chemical, FUJIFILM Wako chemicals or Sigma-Aldrich."
3. We added supplementary materials of obtained compounds including the HMBC and nOe experiments of compound 3a.
4. With reference to Baran's example (Scheme 1, eq. 1), we planned to electrophilically activate the nitrogen atom of aniline to carry out a continuous cyclization reaction (eq. 2). Therefore, Z alkene was selected as the substrate for the cyclization reaction. This explanation is on page 2, lines 43-45.
Round 2
Reviewer 3 Report
My comments were all taken into account and included by the authors in the text of the manuscript and supplementary.